# Microstructure and Wear Resistance of High-Chromium Cast Iron with Multicomponent Carbide Coating via Laser Cladding

Chao Chen [1], Junfa Wang [1,*], Yiyuan Ge [2], Minghui Zhuang [2] and Zheng Ma [2]

1   School of Materials Science and Engineering, Jiamusi University, Jiamusi 154007, China; chenchao7911@126.com
2   School of Mechanical Engineering, Jiamusi University, Jiamusi 154007, China; geyiyuan@jmsu.edu.cn (Y.G.); jmsdxzmh@163.com (M.Z.); jmsdxmz@163.com (Z.M.)
*   Correspondence: wangjunf2934@sina.com

**Abstract:** High-chromium cast iron (HCCI) coatings with multicomponent carbides were prepared on low-alloy steel substrates using a laser cladding technique in this work. The microstructure and wear resistance of the coatings were characterized via optical microscopy, scanning electron microscopy, transmission electron microscopy, energy-dispersive spectroscopy, X-ray diffraction and block-on-ring wear testing. Multicomponent carbides (Ti, Nb, Mo, W, V)C with an FCC structure and multicomponent compounds (Nb, Mo, W, V) (B,C) with an FCC structure were found in the microstructures of coatings after multielement doping. In addition, (Cr, Mo, W, V)$_{23}$C$_6$ compounds could be obtained by heat treatment. These multicomponent compounds were beneficial for obtaining coatings with an excellent hardness (60 HRC) and high wear resistance. This multielement doping method provides an effective modified method for preparing high-wear-resistance laser cladding coatings.

**Keywords:** multicomponent carbides; HCCI; laser cladding coatings; microstructure

## 1. Introduction

With excellent wear resistance, high hardness and low cost, high-chromium cast iron (HCCI) coatings have been used frequently in applications such as mining, hammers, kitchen knives, crushers and liner plates [1–4]. In general, high-chromium cast iron contains three elements, namely Fe, Cr and C, and Fe is the basic element. C plays the role of solid solution strengthening, and the content of Cr and C determines the hardness, corrosion resistance and wear resistance of high-chromium cast iron [5,6]. High-chromium cast iron contains a large amount of M$_7$C$_3$ carbides, which are the foundation of high wear resistance [7,8]. However, the coarse size of M$_7$C$_3$ carbides reduces the impact toughness of the coating, which decreases the wear resistance. After doping with some alloy elements, small compounds (such as MC, MB and M$_2$B; M = Cr, Mo, W, Nb, Ti, etc.) could be formed in the matrix to improve the impact toughness [9–12]. In addition, appropriate heat treatment could form superfine M$_{23}$C$_6$ carbides, further improving the hardness and wear resistance [13].

Liu et al. [14] added a multicomponent V-Fe-Ti-Nb-C-Zr-B alloy (VFC) to HCCI, showing a significant synergistic solution-strengthening effect. They found that the added V-Ti-Nb-B was dissolved in M$_7$C$_3$ carbide, forming (Cr, Fe, V, Ti, Nb)$_7$(C, B)$_3$ alloy carbides. Zhu et al. [15] studied the microstructure and hardness of ICCI roll materials with different contents of (NbTi)C particles and observed that (NbTi)C particles were present in granular, rod and polygonal forms. Wang et al. [16] studied the effects of vanadium on the microstructure and wear resistance of a high-chromium cast iron hardfacing layer using electroslag surfacing and found V can replace a part of the Cr in M$_7$C$_3$ to form (Cr$_{4.4–4.7}$Fe$_{2.1–2.3}$V$_{0.2–0.5}$)C$_3$-type carbides. In previous studies [17], we also found

$(Fe_{3.27}Cr_{2.99}W_{0.74})\,C_3$ carbides in high-chromium cast iron coating. Purba et al. [18] evaluated the relationship between the microstructure and three-body abrasive wear behavior of high-chromium-based (V, Mo, W and Co) multicomponent white cast iron materials. These studies showed that multielement doping can form multicomponent compounds, effectively improving the wear resistance of high-chromium cast iron. In addition, $Cr_3C_2$ carbides could also enhance the wear resistance of the substrate [19,20]. However, there have been few studies on multicomponent carbide coating via laser cladding.

Laser cladding technology [21–23] is a new green reconstruction technology with high economic benefits and excellent quality. Zheng et al. [21] prepared C22, Hastelloy C22 alloy and Ti–6Al–4V alloy coatings via laser cladding technology. Liu et al. [22] prepared Ni–XCr–Mo coatings via laser cladding technology. Zheng et al. [23] prepared Ni-Cr-Mo alloy coatings via laser cladding technology. From this, it can be seen that the laser cladding technology has matured. In addition, $(Cr, Fe)_{23}C_6$ carbide coatings could improve hardness [24].

In this work, two multicomponent V-Fe-Ti-Nb-C-Mo-W-Cr-B high-chromium cast iron coatings were prepared using laser cladding technology, and the microstructure and wear resistance were investigated. This coating is expected to be applied in the repair of digger teeth, hammers, kitchen knives, liner plates, crushers, loose soil plow shovels, automobile parts, etc.

## 2. Experimental Details

In this experiment, a low-alloy steel with a yield strength of 235 MPa and a hardness of 150 HB was selected as the substrate, and the chemical composition is shown in Table 1. Powdered raw materials containing ferrotitanium (66 wt% Ti and 34 wt% Fe), ferrotungsten (70 wt% W and 34 wt% Fe), ferroniobium(65 wt% Nb and 35 wt% Fe), ferromolybdenum (60 wt% Mo and 40 wt% Fe) and ferrovanadium (51 wt% V and 49 wt% Fe) were mixed into high-chromium cast iron powder. These powders were all from Hebei Wenlun Metal Materials Co., Ltd. Handan, China, and the particle size was 40–120 μm. The cladding layer was prepared using an IPG fiber laser system (YLS-6000 IPG Photonics Corporation, Oxford, MA, USA) [25] with a laser beam size of 5 × 5 mm, a laser power of 3000 W, a scanning speed of 4 mm/s, a powder feeding rate of 15 g/min and a 15 L/min flow of high-purity argon shielding gas. Two different coatings were prepared by laser cladding. After spectral testing, their chemical composition was determined, as shown in Table 2.

**Table 1.** The chemical composition of the substrate (wt%).

| C | Si | Mn | P | S | Fe |
|---|----|----|---|---|-----|
| 0.17 | 0.31 | 0.51 | 0.02 | 0.02 | bal |

**Table 2.** The chemical composition of the cladding layer (wt%).

| Samples | Cr | C | Ti | B | W | Nb | V | Mo | S | Fe |
|---------|------|------|------|------|------|------|------|------|-------|-----|
| Coating A | 20.4 | 3.05 | 1.95 | | 1.88 | 2.03 | 1.86 | 1.95 | 0.025 | bal |
| Coating B | 19.8 | 3.01 | | 1.96 | 1.92 | 2.01 | 1.90 | 1.98 | 0.028 | bal |

The heat treatment processes for Coating A and Coating B were as follows: The samples were initially austenitized at 1000 °C for 100 min and then rapidly transferred into air. For Coating A, the temperatures of 900 °C, 950 °C, 1000 °C and 1050 °C were selected to perform heat treatment, and the effect of heat treatment temperature on the microstructure and properties of coatings was analyzed.

Material characterization was performed on two critical coatings, namely Coating A and Coating B, to evaluate their mechanical properties and wear resistance. The microstructure of each sample was inspected under an OLYMP USBX51 optical microscope (OM, Olympus, Tokyo, Japan), a JSM-6510 scanning electron microscope (SEM, JEOL,

Tokyo, Japan), and a JEM-2100F transmission electron microscope and with the use of energy-dispersive spectroscopy (EDS, Oxford Instruments, Oxford, London, UK). The phase structure of the two coatings before and after heat treatment was determined using the X-ray diffraction (XRD, Tokyo, Japan) method. The wear resistance test of coatings was performed using an MM-200 block-on-ring wear testing machine (Baohan, Jinan, China).

## 3. Results and Discussion

### 3.1. Surface Composition

The surface composition of the two coatings was analyzed via X-ray diffraction, and Figure 1 shows the XRD patterns results of the two coatings before and after heat treatment (1000 °C for 100 min). For Coating A, martensite, (Ti, Nb)C, WC, $(Cr, Fe)_7C_3$ and $Mo_2B$ phases were found. For Coating B, martensite, NbC, WC, $(Cr, Fe)_7C_3$ and $Mo_2B$ phases were found. Both samples of Coating A showed diffraction peaks at 40.91°, 59.01° and 70.60°, consistent with the standard (Ti, Nb)C diffraction spectra (PDF#47-1418), corresponding to the crystallographic indices of (200), (220) and (311), respectively. Both samples of Coating B showed diffraction peaks at 40.03°, consistent with the standard NbC diffraction spectra (PDF#38-1364), corresponding to the crystallographic index of (200). It was noted that the content of martensite for the two coatings increased after heat treatment.

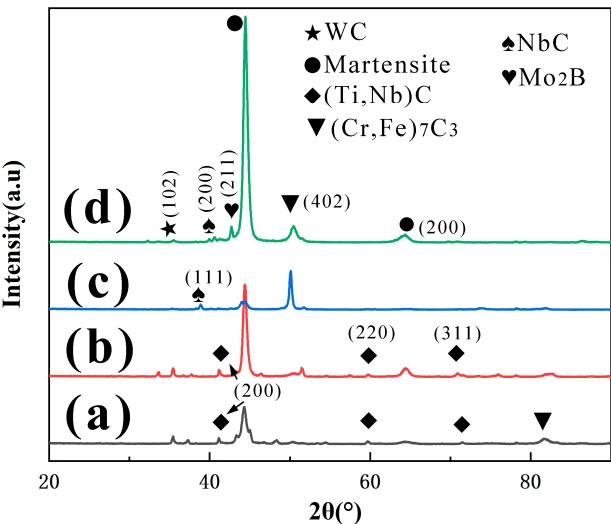

**Figure 1.** XRD patterns of two coating samples: (a) Coating A; (b) Coating A after heat treatment; (c) Coating B; (d) Coating B after heat treatment.

### 3.2. Microstructure before Heat Treatment

In order to better describe the effect of Ti and Nb on high-chromium cast iron coatings, the microstructure of two coatings before and after erosion was observed using an optical microscope as shown in Figure 2. Due to differences in contrast, dot-like MC carbides could be observed in non-corroded Coating A, and the volume fraction of MC carbides was about 5.6%, as shown in Figure 1a. Unfortunately, MC carbides cannot be found in non-corroded Coating A using OM. All the volume fractions of carbides or borides (MC, $(Cr, Fe)_7C_3$ and $M_2B$; M = Cr, Fe, Mo, W, Nb and Ti) were calculated using Image-pro software. The volume fractions in Coating A and Coating B were 35% and 25%. It follows then that the amounts of carbides and borides in the coating increased after the addition of Ti.

Figure 3 shows the microstructure of high-chromium cast iron Coating A and the elemental mapping of elements Nb, Ti, Mo, Fe, Cr, C, W and V. In Figure 3a, the microstructure of high-chromium cast iron Coating A includes eutectic $M_7C_3$ carbides, MC carbides, Mo-rich carbides and an $\alpha$-Fe matrix. In Figure 3b, some compounds are slightly brighter than $M_7C_3$ carbides, MC carbides and an $\alpha$-Fe matrix, such as point 3 and point 4. The EDS spectra results show the compounds of point 3 and point 4 contain a large amount of Mo. Significantly, MC carbide, with its multiplicity of elements Nb, Ti, Mo, W and V, is a

particularly multicomponent carbide (see Figure 3c). In addition, $M_7C_3$ carbides include a large amount of Cr, which is the key to ensuring the wear resistance of a high-chromium cast iron coating. The hardness of Coating A was 42 HRC before heat treatment.

Figure 4 shows the microstructure of high-chromium cast iron Coating B and the elemental mapping of elements Nb, B, Mo, Fe, Cr, C, W and V. In Figure 4a, the microstructure of high-chromium cast iron Coating A includes eutectic $M_7C_3$ carbides, an Nb-rich compound and an $\alpha$-Fe matrix. In Figure 4b, it is not hard to see that the Nb-rich compounds were slightly brighter than the $M_7C_3$ carbides and $\alpha$-Fe matrix; examples include point 1 and point 2. Unlike Coating A, in Coating B, the Nb-rich compound does not contain Ti but contains B, as shown in Figure 4c. Similarly, the Nb-rich compound, with its multiplicity of elements Nb, B, Mo, W and V, is a particularly multicomponent compound. The hardness of Coating B was 40 HRC before heat treatment.

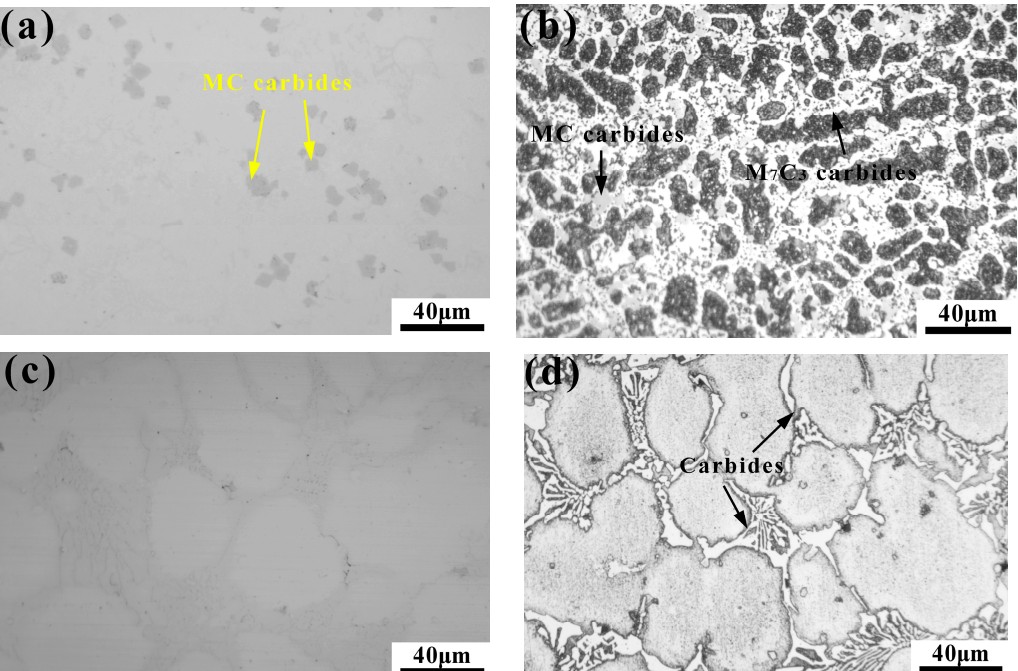

**Figure 2.** OM: (**a**) Coating A before heat treatment, non-corroded; (**b**) Coating A before heat treatment, corroded by the aqua regia; (**c**) Coating B before heat treatment, non-corroded; (**d**) Coating B before heat treatment, corroded by the aqua regia.

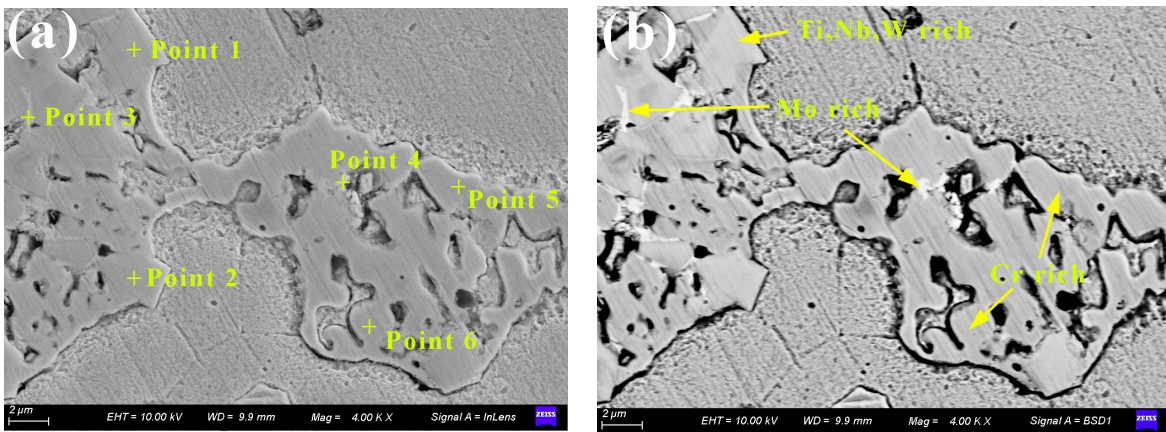

**Figure 3.** *Cont.*

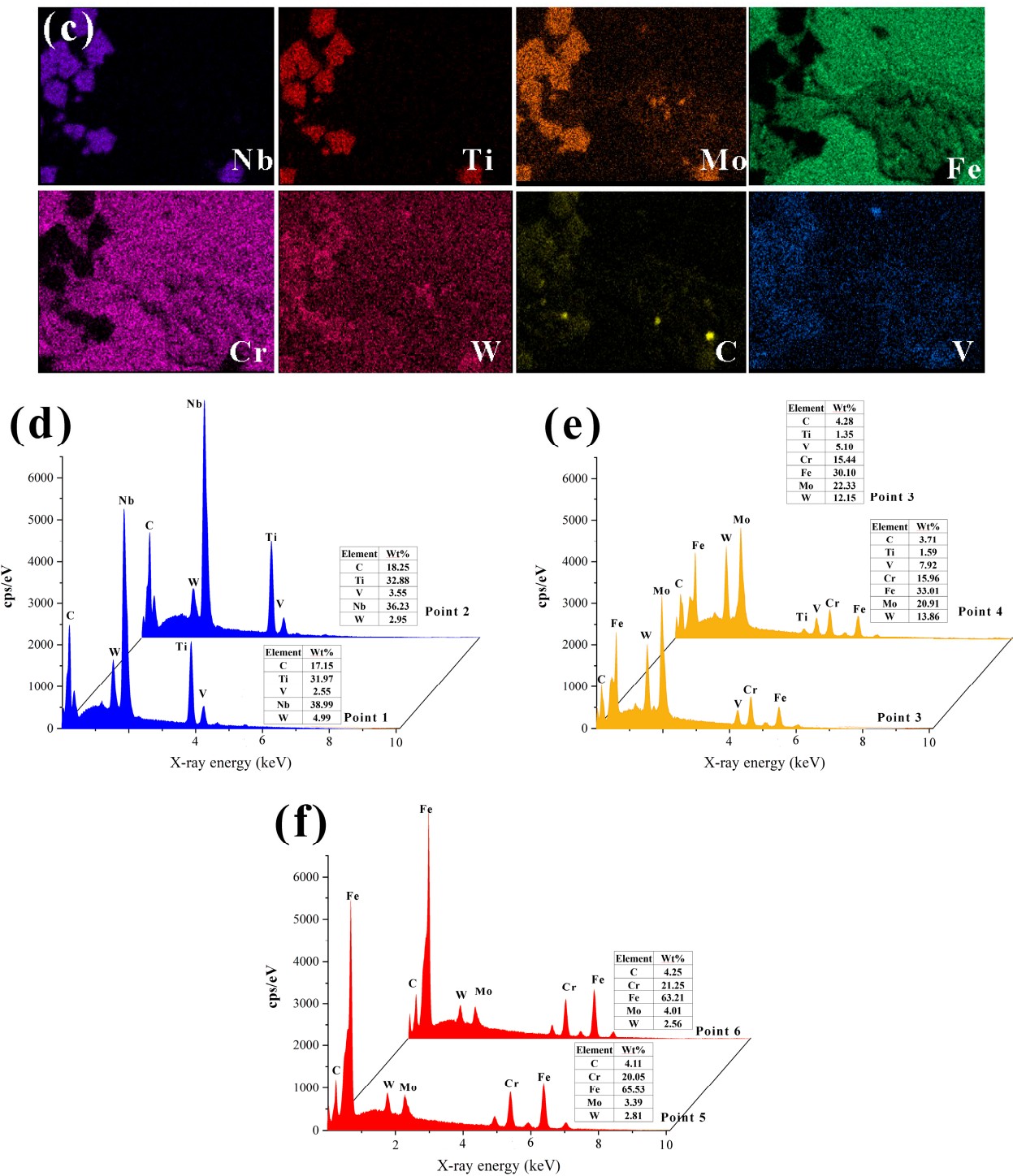

**Figure 3.** SEM and EDS of Coating A: (**a**) SEM image; (**b**) BSE image; (**c**) elemental mapping of (**a**); (**d**) EDS spectra attachment of point 1 and point 2; (**e**) EDS spectra attachment of point 3 and point 4; (**f**) EDS spectra attachment of point 5 and point 6.

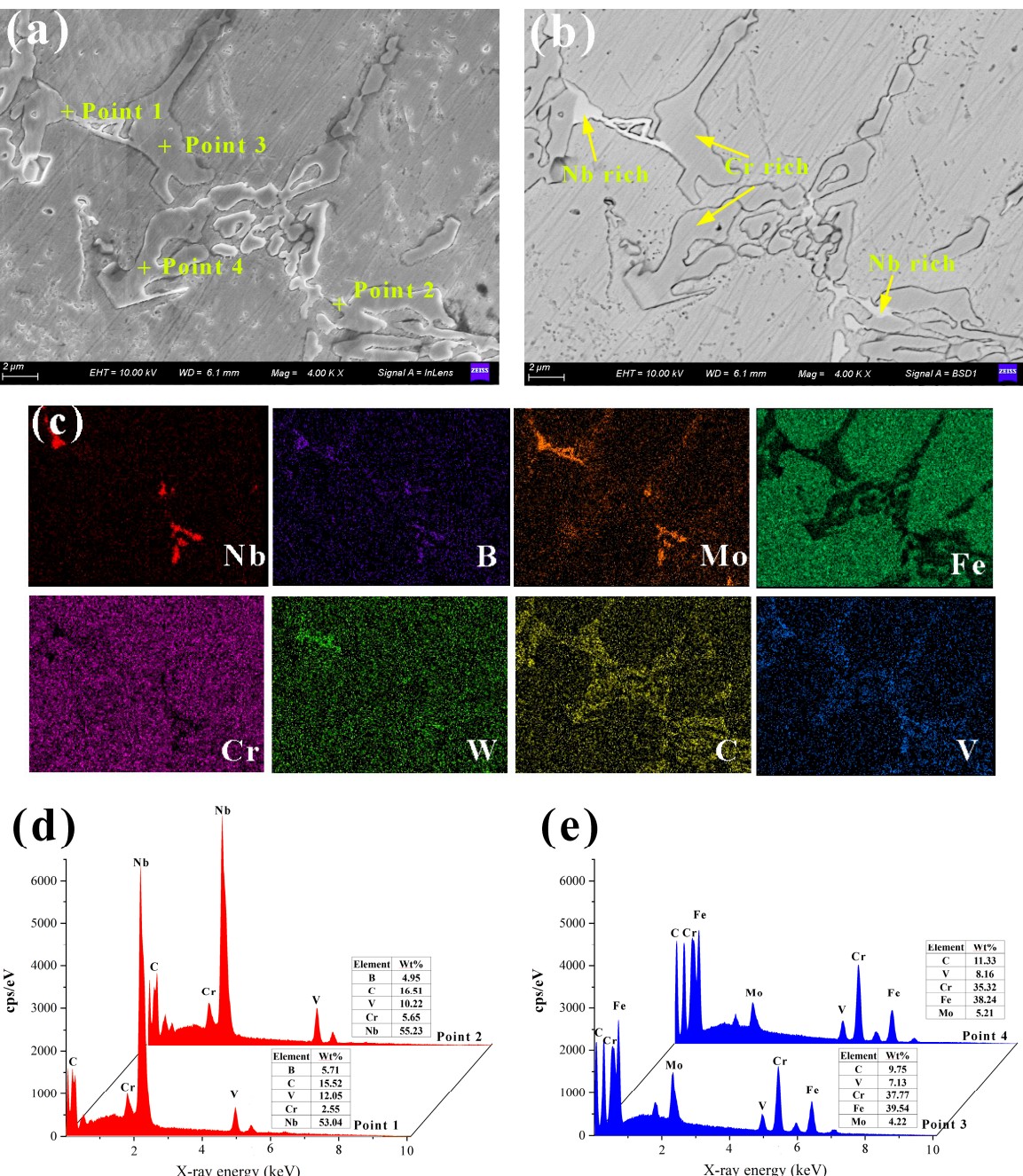

**Figure 4.** SEM and EDS of Coating B: (**a**) SEM image; (**b**) BSE image; (**c**) elemental mapping of (**a**); (**d**) EDS spectra attachment of point 1 and point 2; (**e**) EDS spectra attachment of point 3 and point 4.

In order to further clarify the lattice types of multicomponent compounds in Coating A and Coating B, the microstructure was analyzed using transmission electron microscopy (TEM). Figures 5 and 6 show the bright-field TEM image, corresponding selected area diffraction patterns (SADPs) of multicomponent compounds and elemental mapping of Coating A and Coating B, respectively. The bright-field TEM image and SADP results in Figure 5 show that the multicomponent carbides are (Ti, Nb)C with the enrichment of a large number of alloy elements (Mo, W and V). In other words, the multicomponent carbides are (Ti, Nb, Mo, W, V)C carbides. The calculated results based on the SADPs show that the structure of the multicomponent carbides is an FCC structure with a lattice constant of a = b = c = 0.448 nm. Significantly, the bright-field TEM image and SADP results in Figure 6 show that the structure of multicomponent compounds is also an FCC structure.

However, the multicomponent compounds of Coating B contain Nb, Mo, W, V and B. The lattice constant of multicomponent compounds is a = b = c = 0.451 nm. Obviously, the multicomponent compounds should be (Nb, Mo, W, V) (B,C) compounds. M(B,C) is a high-melting-point compound and crystallized first. $M_7C_3$ is a pre-eutectic carbide or eutectic carbide. Therefore, the crystallization order of various compounds should be as follows: M(B,C) compounds forming first, then Mo-rich carbides forming, and finally $M_7C_3$ compounds forming.

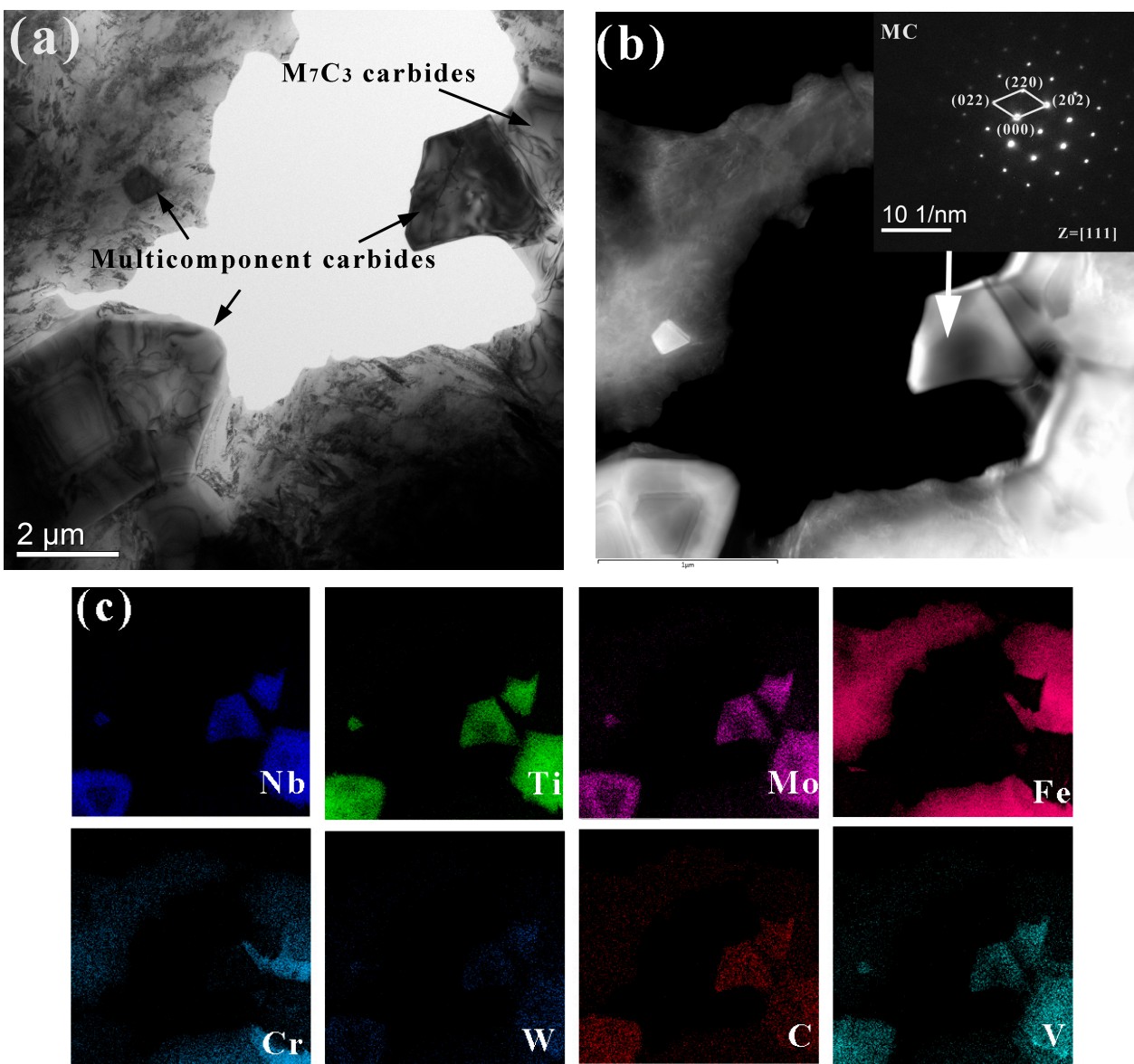

**Figure 5.** TEM of Coating A: (**a**) bright-field TEM image; (**b**) SADPs of multicomponent carbides; (**c**) elemental mapping of (**b**).

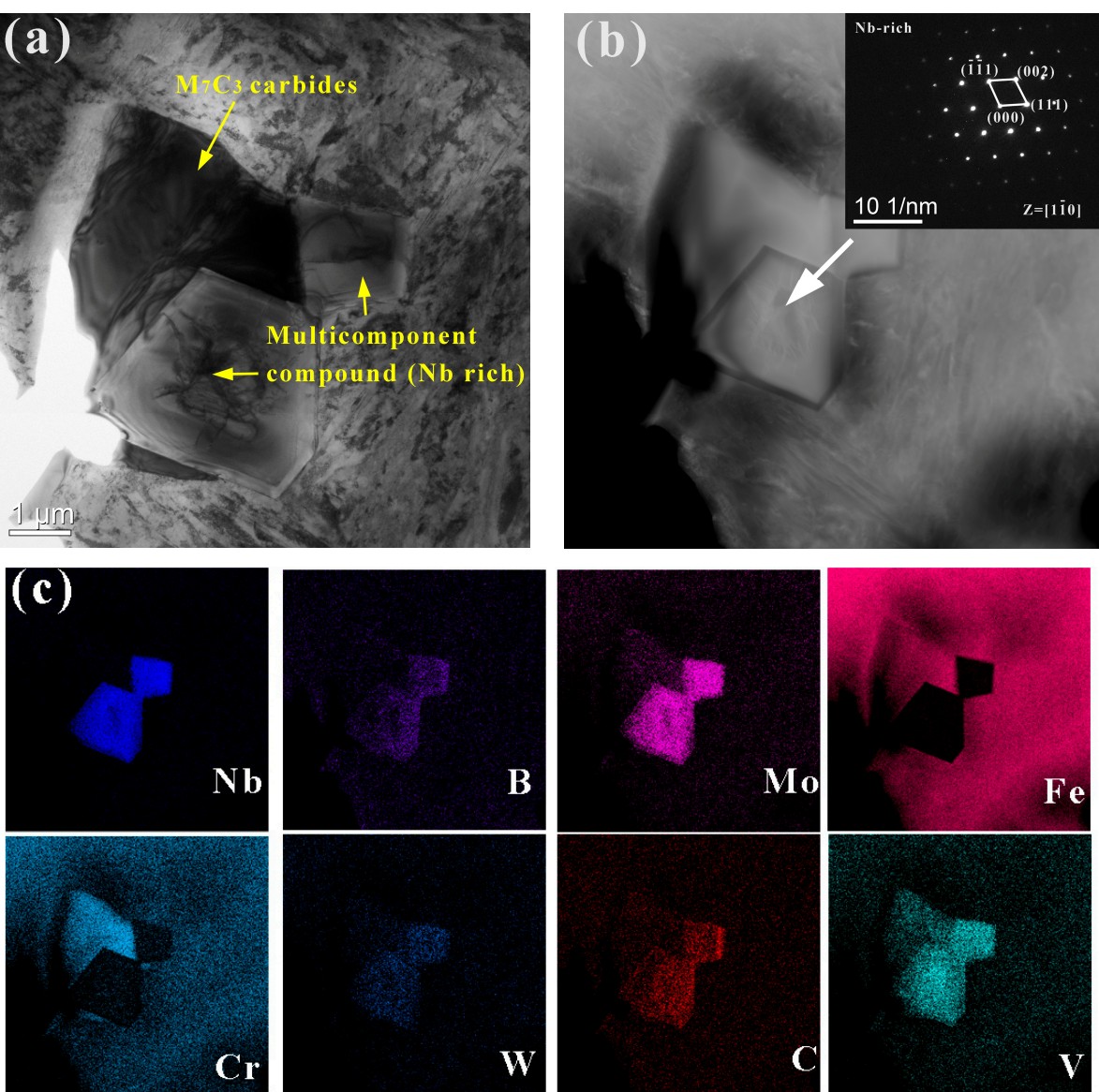

**Figure 6.** TEM of Coating B: (**a**) bright-field TEM image; (**b**) SADPs of multicomponent compounds; (**c**) elemental mapping of (**b**).

*3.3. Microstructure after Heat Treatment*

Figures 7 and 8 show OM images and SEM images of Coating A austenitized at different temperatures, respectively. The multicomponent carbides can be clearly observed in the non-corroded OM image. It is not difficult to see that the morphology and size of carbides do not change after heat treatment at different temperatures. In order to further analyze the changes in carbides, scanning electron microscopy was used to observe the microstructure of Coating A at different temperatures. As the temperature increases, the volume fraction of multicomponent carbides gradually decreases, and a certain amount of fine particulate appears in the matrix. To reflect the quantity of carbides more clearly, the volume fraction of carbides was calculated using Image pro software in this work, as shown in Figure 9. Before heat treatment, the volume fraction of carbides was 50.2%. With the increase in temperature from 900 °C to 1050 °C, the volume fraction of carbides decreased from 48.5% to 36.2%.

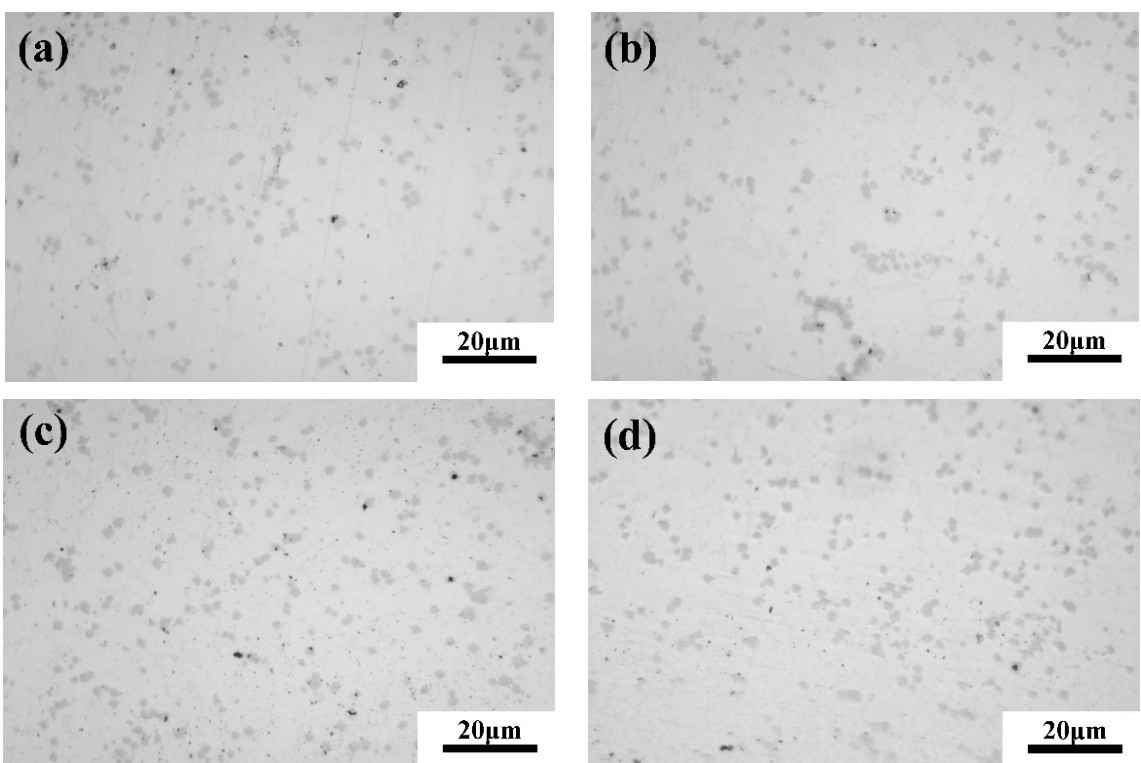

**Figure 7.** OM after heat treatment with different temperatures, non-corroded: (**a**) 900 °C; (**b**) 950 °C; (**c**) 1000 °C; (**d**) 1050 °C.

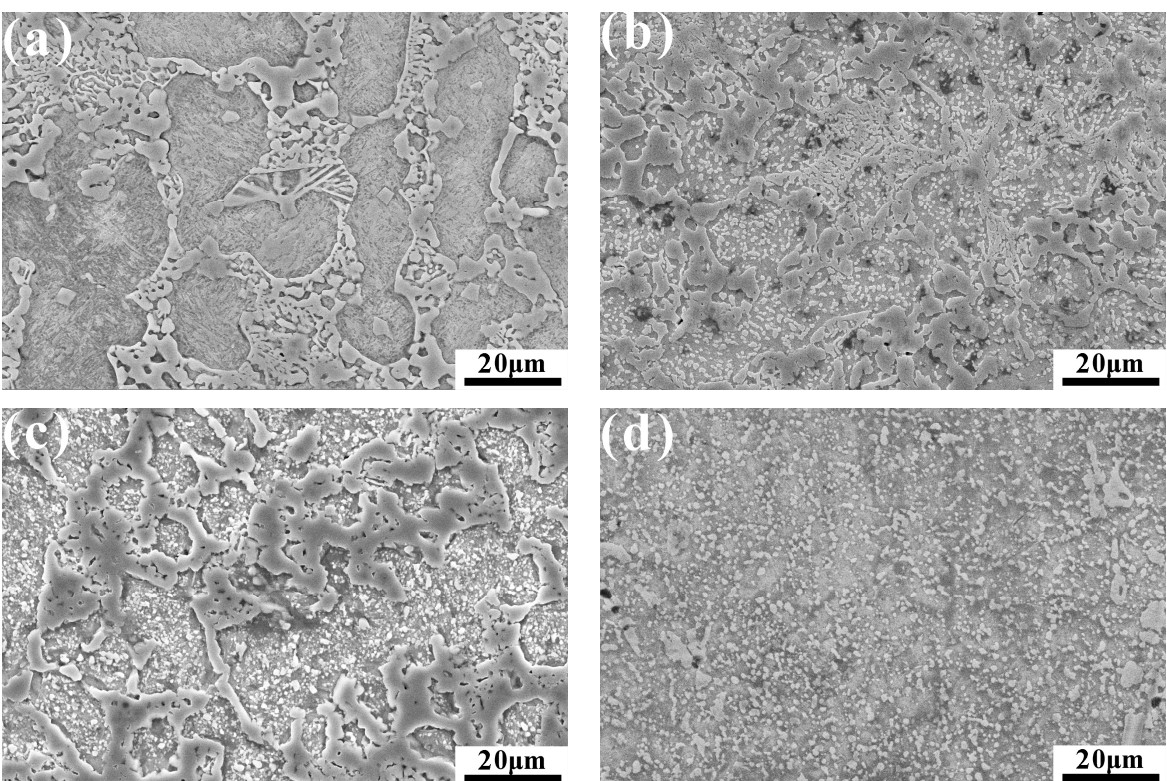

**Figure 8.** SEM after heat treatment with different temperatures: (**a**) 900 °C; (**b**) 950 °C; (**c**) 1000 °C; (**d**) 1050 °C.

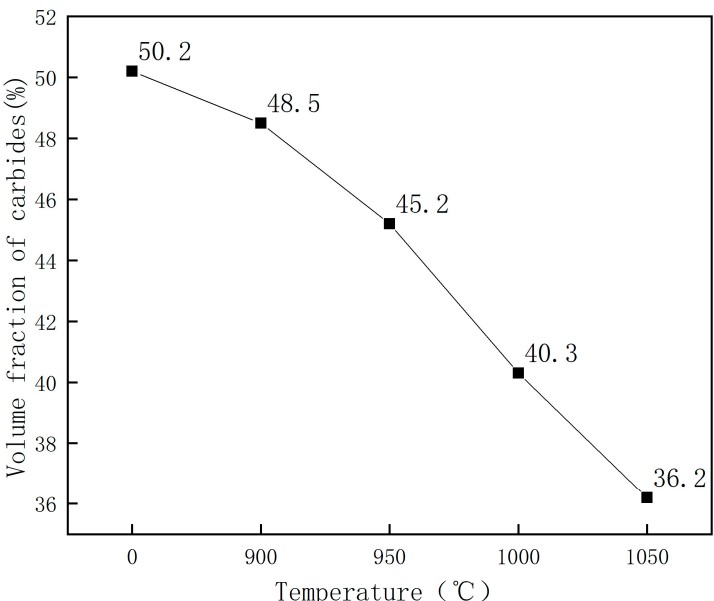

**Figure 9.** The volume fraction of multicomponent carbides in Coating A.

In order to analyze the fine particulates in the microstructure of Coating A treated at 1050 °C, the microstructure was analyzed by transmission electron microscopy and energy-dispersive spectroscopy. Figure 10 shows bright-field TEM images, high-resolution transmission electron microscopy (HRTEM) images and elemental mapping of Coating A after heat treatment at 1050 °C. It is not hard to see that the fine particulates contain Cr, Mo, W and V, and the lattice constant of the fine particulates is a = b = c = 1.057 nm. Obviously, the fine particulates should be (Cr, Mo, W, V)$_{23}$C$_6$ compounds.

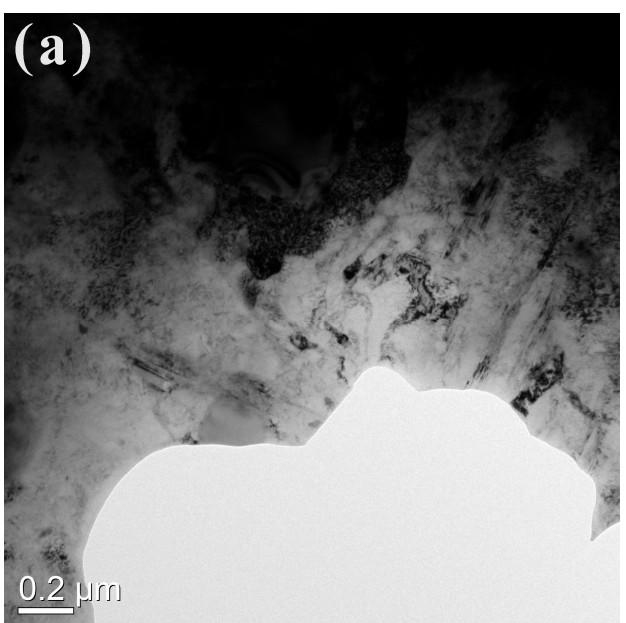
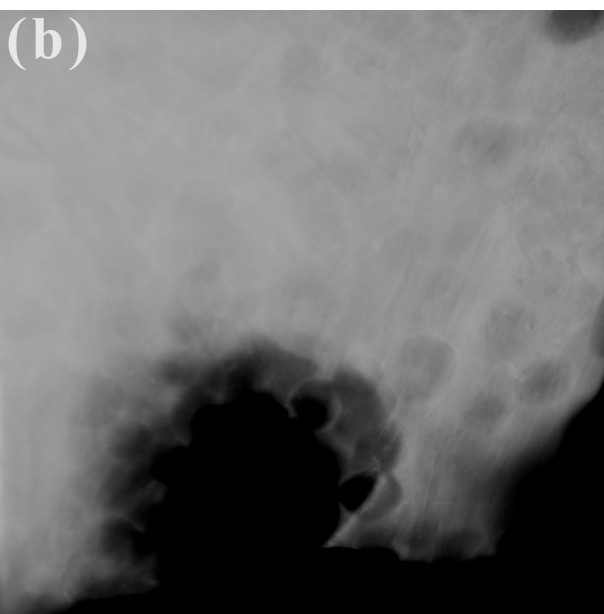

**Figure 10.** *Cont.*

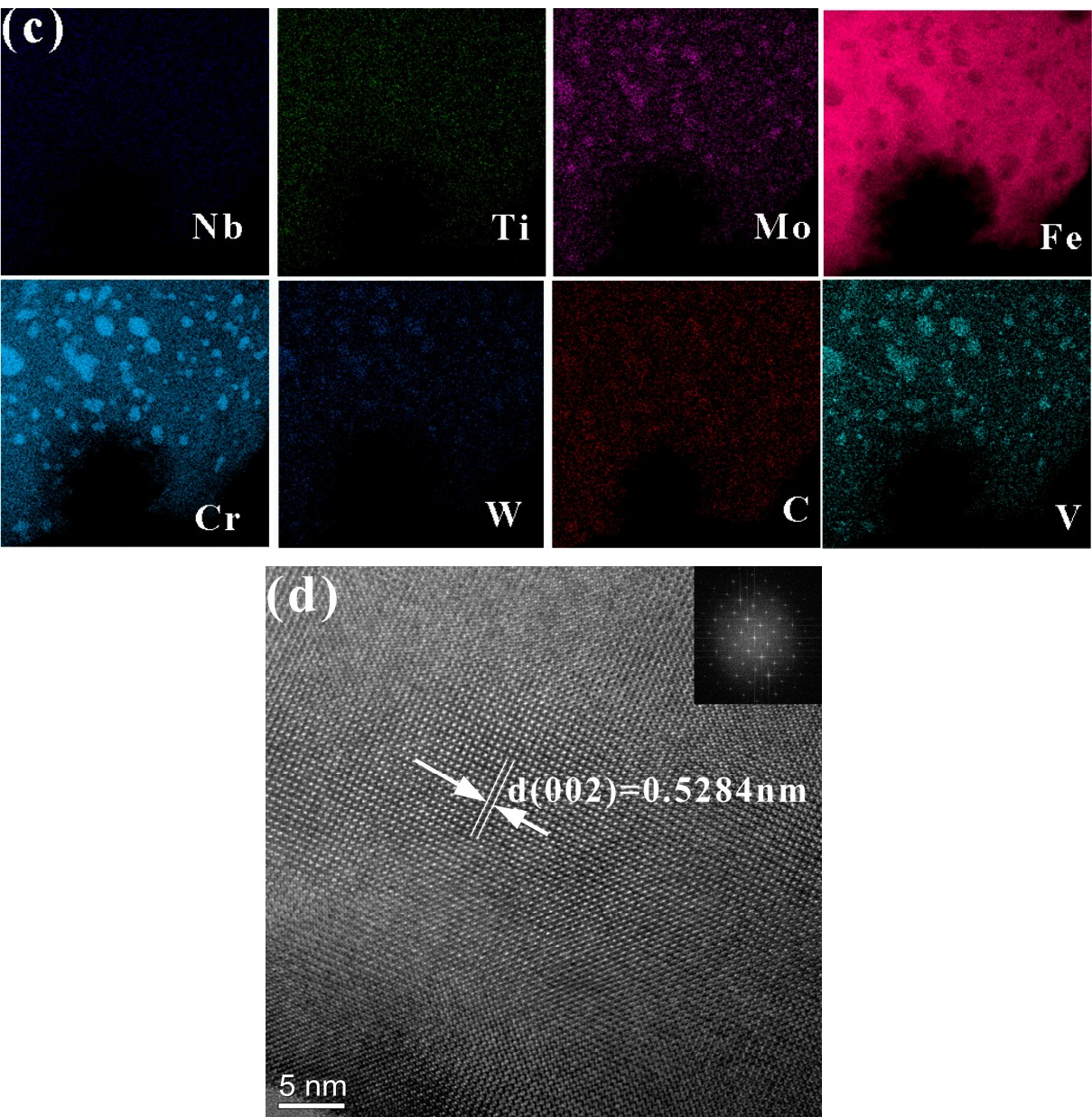

**Figure 10.** TEM of Coating A after heat treatment at 1050 °C: (**a**) bright-field TEM image; (**b**) TEM image; (**c**) elemental mapping of (**b**); (**d**) high-resolution transmission electron microscopy image of fine particulate.

Figure 11 shows an SEM image and BSE image of Coating B at high magnification and low magnification after heat treatment. After heat treatment of 1000 °C, the changes in the size and quantity of the multicomponent compounds (Nb, Mo, W, V) (B,C) were small. It should be noted that $M_7C_3$ carbides were reduced after heat treatment, and fine particulates were found in the matrix. These fine particulates have a square shape and a size of approximately 300 nm, as shown in Figure 11c,d.

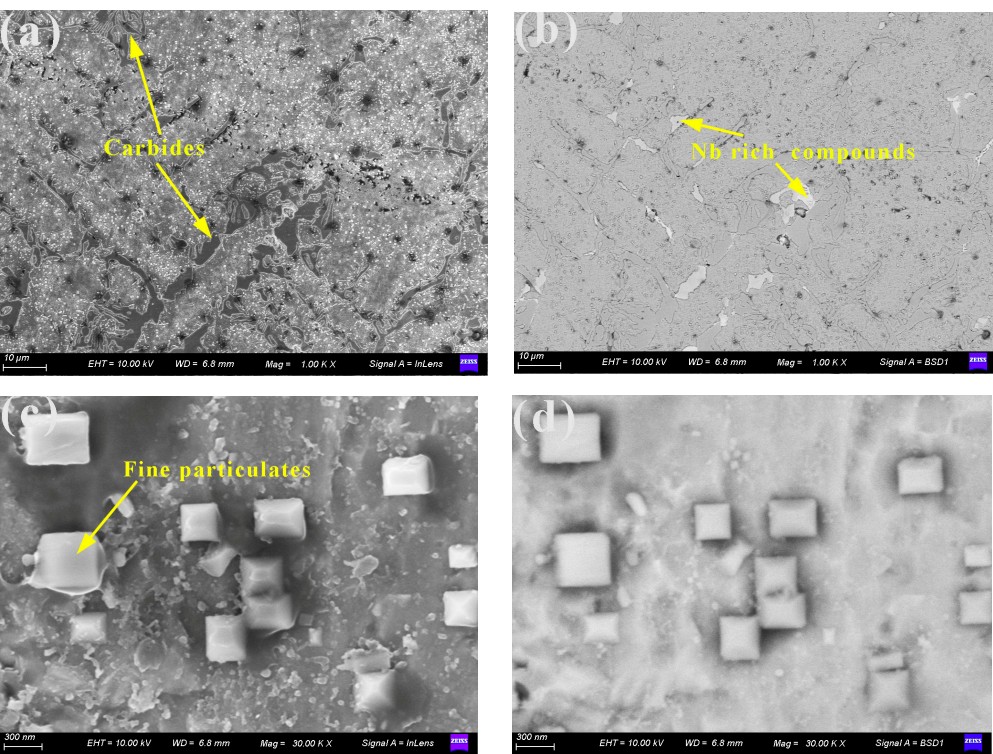

**Figure 11.** SEM and BSE of Coating B after heat treatment: (**a**) SEM image; (**b**) BSE image; (**c**) SEM image at high magnification; (**d**) BSE image at high magnification.

### 3.4. Properties of Coatings

Figure 12 shows the hardness of Coating A and Coating B after heat treatment. Due to the decrease in $M_7C_3$ carbides and the increase in fine particulates, the hardness of Coating A gradually increased with the heat treatment temperature. It is worth noting that under the same heat treatment temperature, the hardness of Coating B is equivalent to that of Coating A. In general, the addition of element B can increase the hardness of the coating. However, the addition of element B did not increase the hardness in this study, which may be due to the decrease in Ti.

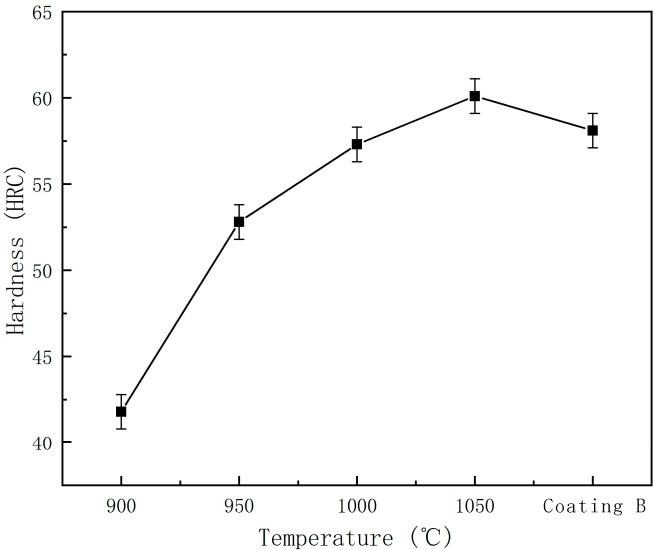

**Figure 12.** Hardness of Coating A and Coating B after heat treatment.

A wear test was carried out using the block-on-ring wear testing machine with a load of 200 N, a rotational speed of 300 r/min and a wear time of 240 min. Figure 13 shows the wear loss of Coating A and Coating B after heat treatment; the more mass loss, the better the wear resistance. The result showed that the wear resistance of Coating A gradually increased with the heat treatment temperature, which is consistent with the trend of hardness changes. From this, it can be seen that the wear resistance gradually increased with the hardness. It is worth noting that under the same heat treatment temperature, the wear resistance of Coating B is equivalent to that of Coating A. Although there are many factors that affect wear resistance, such as hardness, impact toughness and second phase, hardness greatly affects wear resistance. Therefore, the higher the hardness, the better the wear resistance in this study.

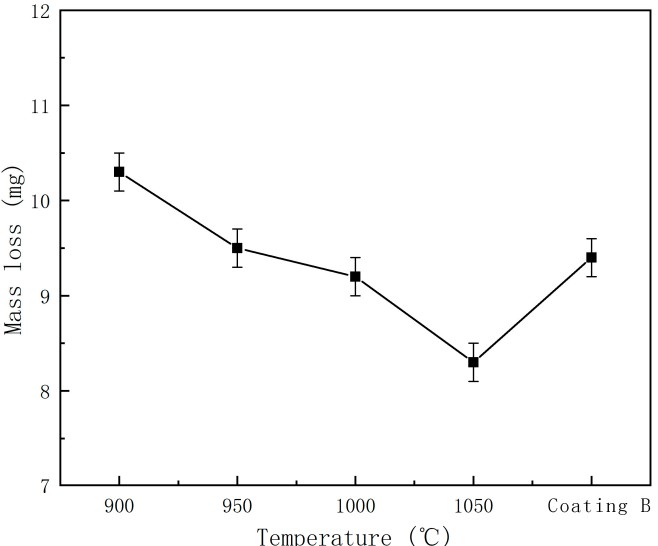

**Figure 13.** Wear resistance of Coating A and Coating B after heat treatment.

Scanning electron microscopy was used to observe the worn surface of coating samples, as shown in Figure 14. It is not hard to see that a large extent of spalling and a number of grooves are presented on the worn surface, which is typically adhesion wear and abrasive wear. During the wear process, high-hardness carbides, oxides and nitrides appeared on the sample surface, which led to the formation of grooves. The differences in worn surface morphology between the two coatings are very small, indicating that the coatings have similar wear resistance under the same heat treatment process conditions.

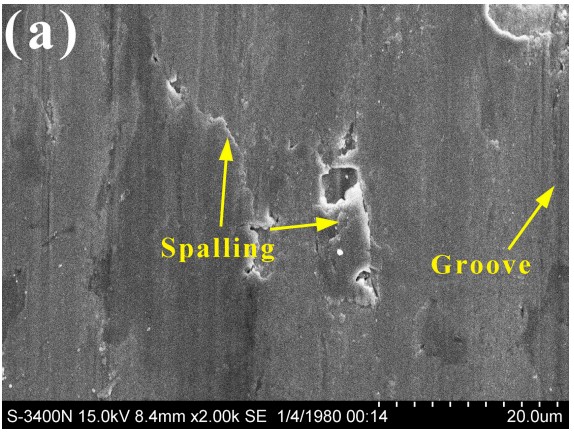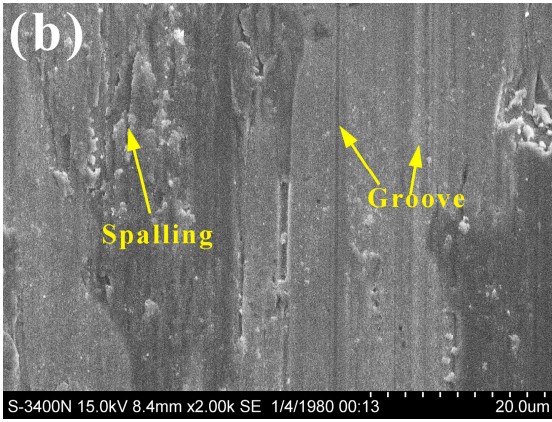

**Figure 14.** SEM images of the worn surfaces after heat treatment at 1000 °C for 100 min: (**a**) Coating A; (**b**) Coating B.

## 4. Conclusions

In this work, multiple elements were added to high-chromium cast iron laser cladding coatings to investigate their microstructure and properties experimentally.

(a) Multicomponent carbides (Ti, Nb, Mo, W, V)C with an FCC structure and multicomponent compounds (Nb, Mo, W, V) (B,C) with an FCC structure in high-chromium cast iron coating could be obtained via laser cladding.

(b) After heat treatment, (Cr, Mo, W, V)$_{23}$C$_6$ compounds appeared in the matrix of the coating, which increased hardness and wear resistance.

(c) With the increase in heat treatment temperature, the hardness and the wear resistance of the coating gradually increased.

## 5. Novelty and Application

After multielement doping, (Ti, Nb, Mo, W, V)C, (Cr, Mo, W, V)$_{23}$C$_6$, (Cr, Fe)$_7$C$_3$ and (Nb, Mo, W, V) (B,C) compounds could be obtained to improve the hardness and wear resistance of high-chromium cast iron laser cladding coating. This coating is expected to be applied in the repair of digger teeth, hammers, kitchen knives, liner plates, crushers, loose soil plow shovels, automobile parts, etc.

**Author Contributions:** C.C.: Investigating, SEM Analysis, Writing, Review, Editing. J.W.: Supervision, Funding Acquisition, Writing, Review, Editing. Y.G.: Investigation, Experiments, Data Curation. M.Z.: TEM Analysis, Writing, Review, Editing. Z.M.: Property Analysis, Writing, Review, Editing. All authors have read and agreed to the published version of the manuscript.

**Funding:** This research is supported by the National Natural Science Foundation of China under grant 51875252, the Natural Science Foundation of Heilongjiang Province (LH2020E026) and the Basic Scientific Research Business Fee Project of Heilongjiang Provincial Department of Education (2020-KYYWF-0266).

**Institutional Review Board Statement:** Not applicable.

**Informed Consent Statement:** Not applicable.

**Data Availability Statement:** The datasets generated during and/or analyzed during the current study are available from the corresponding author on reasonable request.

**Conflicts of Interest:** The authors declare no conflict of interest.

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
