# Peer review of "Microstructure and Wear Resistance of High-Chromium Cast Iron with Multicomponent Carbide Coating via Laser Cladding"

_coatings, doi:10.3390/coatings13081474_

Round 1
Reviewer 1 Report
Considering that the authors have done a very good characterization in this research. Unfortunately, the article has fundamental problems and cannot be accepted in this journal. It is necessary for the authors to solve these problems.
-The main goal of this research is to investigate the wear properties. However, a proper investigation in terms of the relationship between hardness and wear, types of wear mechanisms at different temperatures, and times of heat treatment has not been done. You should add macro and micro images after the wear test.
-There are many grammatical and spelling errors in the text, so it should be revised from the point of view of language and writing. The text of the article is very poorly written and beginner. As:
Page 1, line 8: HCCL
Page 1, line 13: The word Compounds is used twice.
-The abstract needs to be substantially revised. The introduction needs to be fundamentally modified. The aim and innovation of the article is not clear.
Moderate editing of English language required.
Author Response
The authors appreciate the helpful comments provided by the referee. The revisions addressing the issues and comments are listed and explained below, in addition to those revisions and corrections from our own considerations. To facilitate the review process, response to each of the issues and comments is presented item by item. Moreover, thank you for your suggestion very much, and we benefit greatly from these suggestions.

Reviewer 2 Report
Review report: Microstructures and wear resistance of high chromium cast iron with multicomponent carbides coating via laser cladding. Work is presented well with good publishing quality and can be accepted after the following corrections:
1. Abstract: Add some quantitative results related to mechanical testing at end of the abstract section.
2. Introduction: In place of citing multiple references, explain the individual work of the author and try to make a bridge between current and previous work. Refer to some recently published work.
https://doi.org/10.1007/s12666-016-0977-6 (Not mine); https://doi.org/10.1007/s12540-020-00705-w (not mine).
3. Novelty and application: Add a separate section for novelty and application of work. Also add clear discussion related to use of different coating in experimental section.
4. Add a detail of experimental set up instead of a schematic image. Also add the parameters and provide the mechanical properties of the used material.
5. Add image of the plate after coating.
6. Add composition of substrate in a table. Also add the SEM image of the coating powder.
7. In XRD analysis, add the quanitative results of phase analysis.
8. In EDS, add the composition at each point selected for analysis.
9. Discuss the formation of the various phases at different stages.
10. Role of heat treatment is not clear.
11. How was the volume fraction of the carbides measured?
12. In the hardness plot only information of coating B is presented. Also with an increase in temperature increase in hardness is not discussed. Try to relate it with the microstructure evolution and add a clear discussion.
13. Wear information is also incomplete. Add an image of the wear setup. Wear out surfaces and also detail of the wear mechanism or remove this section.
NA
Author Response
The authors appreciate the helpful revise provided by the reviewer. The revisions addressing the issues and comments are listed and explained below, in addition to those revisions and corrections from our own considerations. The changes in the manuscript are highlighted in red. Both clear manuscript and marked up manuscript are provided. Moreover, thank you for your suggestion very much, and we benefit greatly from these suggestions.

Reviewer 3 Report
The paper “Microstructures and wear resistance of high chromium cast iron with multicomponent carbides coating via laser cladding” is a study about the effect of multicomponent carbides in High Chromium Cast Iron (HCCL) coating on low alloy steel substrates by Laser cladding technique. Hardness and wear are evaluated on two different coatings with different heat treatments. The scientific quality of the papers is acceptable, and the topic is interesting, but not so new. The scientific method appears correct, and the research activities are well programmed.
I suggest to accept the manuscript after minor revision.
1. Please show the values about hardness and wear also in some tables to improve the clarity of results.
2. It is very strange that the coating A and coating B have the same values of hardness and wear. It is probably that are very similar, please shows the values for both coating.
3. The conclusions are too concise, please argue with more details.
Round 2
Reviewer 1 Report
Accept in present form.
Reviewer 2 Report
Accepted
NA